# Glycemic Control after Initiation of Anti-VEGF Treatment for Diabetic Macular Edema

**DOI:** 10.3390/jcm11164659

**Published:** 2022-08-09

**Authors:** Hideyuki Oshima, Yoshihiro Takamura, Takao Hirano, Masahiko Shimura, Masahiko Sugimoto, Teruyo Kida, Takehiro Matsumura, Makoto Gozawa, Yutaka Yamada, Masakazu Morioka, Masaru Inatani

**Affiliations:** 1Department of Ophthalmology, Faculty of Medical Sciences, University of Fukui, Yoshida 910-1193, Fukui, Japan; 2Department of Ophthalmology, Shinshu University School of Medicine, Nagano 390-0802, Matsumoto, Japan; 3Department of Ophthalmology, Tokyo Medical University Hachioji Medical Center, Tokyo 193-0998, Hachioji, Japan; 4Department of Ophthalmology, Mie University Graduate School of Medicine, Tsu 514-8507, Mie, Japan; 5Department of Ophthalmology, Osaka Medical and Pharmaceutical University, Takatsuki 569-8686, Osaka, Japan

**Keywords:** diabetic macular edema, DME, glycemic control, anti-VEGF therapy, medical expenses, hemoglobin A1c, HbA1c

## Abstract

Diabetic macular edema (DME) induces visual disturbance, and intravitreal injections of anti-vascular endothelial growth factor (VEGF) drugs are the accepted first-line treatment. We investigate its impact on glycemic control after starting VEGF treatment for DME on the basis of a questionnaire and changes in hemoglobin A1c (HbA1c). We conducted a retrospective multicenter study analyzing 112 patients with DME who underwent anti-VEGF therapy and their changes in HbA1c over two years. Central retinal thickness and visual acuity significantly improved at three months and throughout the period after initiating therapy (*p* < 0.0001); a significant change in HbA1c was not found. A total of 59.8% of patients became more active in glycemic control through exercise and diet therapy after initiating therapy, resulting in a significantly lower HbA1c at 6 (*p* = 0.0047), 12 (*p* = 0.0003), and 18 (*p* = 0.0117) months compared to patients who did not. HbA1c was significantly lower after 18 months in patients who stated that anti-VEGF drugs were expensive (*p* = 0.0354). The initiation of anti-VEGF therapy for DME affects HbA1c levels in relation to more aggressive glycemic control.

## 1. Introduction

Diabetic retinopathy (DR) is a common microvascular complication of diabetes mellitus (DM), diabetic nephropathy, and diabetic neuropathy [1]. After a prolonged asymptomatic nonproliferative phase, DR progresses to a proliferative phase with neovascularization, leading to vision loss due to vitreous hemorrhage and tractional retinal detachment. Diabetic macular edema (DME), which can occur at any stage of DR, also induces visual disturbance.

The levels of hemoglobin A1c (HbA1c), a glycosylated protein, are not affected by short-term blood glucose concentration; therefore, they represent overall glycemic control over a period of 2–3 months. HbA1c is the gold standard for long-term blood glucose control. Several studies have shown that the occurrence and development of DR is closely related to HbA1c level [2,3]. In addition to drug therapy with insulin and hypoglycemic agents, self-management with dietary restriction and exercise at an appropriate load is important for good glycemic control. Patient education about DM and the motivation for treatment strongly influence self-management behavior and blood glucose control.

The intravitreal injection of antivascular endothelial growth factor (VEGF) drugs is currently the accepted first-line treatment for DME [4,5]. Anti-VEGF agents restore the integrity of the blood retinal barrier and resolve the macular edema. Recent randomized clinical trials demonstrated the efficacy and safety of anti-VEGF therapy for DME, with greater improvement in visual acuity, superior to that of conventional laser photocoagulation [6,7,8,9]. In Japan, two VEGF inhibitors, aflibercept and ranibizumab, have been approved for the treatment of DME by injection into the vitreous cavity. However, multiple injections are required to maintain the therapeutic effect, having an economic burden on the patient and the healthcare system due to the expense of the drug [10,11].

This may increase the risk of diabetic distress in diabetic patients, a stressful emotional state related to difficulties in diabetic management and concerns about diabetic complications [12]. Diabetic distress occurs in 18% to 35% of Type 2 diabetic patients [13], and is associated with poor treatment adherence and glycemic control, leading to increased risk of diabetic complications [14,15,16]. The sudden loss of vision and the bilateral nature of DME, and the financial burden of anti-VEGF treatment may also enhance diabetic distress. Conversely, anatomical and functional improvement may reduce the psychological stress of patients. Indeed, the MERCURY study showed that anti-VEGF therapy reduced anxiety in patients with DME, as visual acuity improved over the course of 2 years. We hypothesized that anti-VEGF treatment for DME may improve the motivation of the patients to adequately control blood glucose. Therefore, we conducted a questionnaire survey of patients with DME who had been on anti-VEGF therapy for 2 years regarding their efforts of glycemic control, and investigated the relationship with changes in HbA1c before and after treatment.

## 2. Materials and Methods

We collected data from five clinical centers throughout Japan. The current study was performed in accordance with the Declaration of Helsinki and approved (IRB number: 20200004; date of approval: 11 January 2022) by the ethics committees of University of Fukui, University of Mie, University of Shinshu, Osaka Medical and Pharmaceutical University Hospital, and Hachioji Medical Center. All patients provided signed informed consent forms. We registered the study with the University Hospital Medical Information Network Clinical Trials Registry (UMIN-CTR) of Japan (ID UMIN000044287). This retrospective, multicenter study consisted of patients with DME who had received intravitreal treatment with anti-VEGF over 2 years between January 2015 and December 2021.

Patients with Type 2 diabetes with a thickening of the central macula, defined as a central retinal thickness (CRT) ≥ 300 μm in the central subfield on the basis of optical coherence tomography (OCT) due to DME, were eligible for this study. Leakage from the capillary retinal vessels and microaneurysms corresponding to macular edema was identified by fluorescein angiography. The main exclusion criteria were: (1) <20 years of age; (2) active intraocular inflammation or infection in either eye; (3) uncontrolled glaucoma in either eye; and (4) a history of stroke. We also excluded patients with retinal diseases other than DR; with severe cataract, corneal diseases, or vitreous hemorrhage resulting in poor-quality OCT image; and with a history of retinal photocoagulation within 6 months before and/or after the initial injection.

A questionnaire survey was conducted after 2 years to investigate the change in motivation for diabetic care caused by anti-VEGF treatment, consisting of four yes or no questions:1.Since starting anti-VEGF therapy, have you become more active in glycemic control through diet and/or exercise therapy?2.Until you were aware of your vision impairment due to DME, did you know that diabetes mellitus can cause visual impairment?3.Have you started to regular visit an ophthalmologist after vision loss?4.Do you think the cost of anti-VEGF drugs is high?

Patients were grouped according to their answers, and changes in HbA1c levels were analyzed in each group. Age and the percentage of medical burden were informed by the patient’s medical records.

The values of HbA1c at the time of initial injection of anti-VEGF drug (baseline), and at 3, 6, 12, 18, and 24 months after the first injection were examined. All patients underwent an examination that included slit-lamp examination, dilated fundus examination, fundus photography, best-corrected visual acuity (BCVA) measurement (Snellen), intraocular pressure (IOP), and CRT measurement using OCT (Cirrus OCT, Carl Zeiss Meditec, Dublin, CA, USA). BCVA measured with a Landolt chart was converted to the logarithm of the minimal angle of resolution (logMAR). Patients were eligible for additional injections after the first injection if CRT was greater than 350 μm [9].

Intravitreal injections were performed in a standard manner by a trained ophthalmologist using 0.4% oxybuprocaine hydrochloride (0.4% benoxyl ophthalmic solution, Santen Co. Ltd., Osaka, Japan) and 2% xylocaine as anesthetic, and povidone iodine for sterilization. The injection volume of ranibizumab (Lucentis; Novoartis Pharma K.K., Tokyo, Japan) and aflibercept (Eylea; Bayer Yakuhin, Ltd., Tokyo, Japan) was 0.5 mg/0.05 mL and 2 mg/0.05 mL, respectively [11].

Statistical analyses were performed using JMP (SAS Institute Inc., Tokyo, Japan). The data are shown as mean ± standard deviation. One-way repeated-measures analysis of variance (RMANOVA) with Greenhouse–Geisser correction was performed to examine whether changes over time were statistically significant. Steel’s multiple-comparison test was used to compare continuous variables within a group. We examined the differences between the groups using the Mann–Whitney test as a nonparametric test. We calculated the effect size, and represented the η^2^ value for RMANOVA, and the r value for Steel’s multiple-comparison and Mann–Whitney U tests. Simple regression analysis was carried out to examine the relationship between the changes in BCVA and HbA1c over the two years. To estimate homoscedasticity, we additionally performed robust regression analysis. The level of statistical significance was set at *p* < 0.05.

## 3. Results

We enrolled 112 patients who were followed for two years and completed the questionnaire. Table 1 shows the baseline characteristics of enrolled patients. The number of anti-VEGF agents injected over the two years was 6.4 ± 4.5 (aflibercept; 3.0 ± 3.7; ranibizumab 2.2 ± 3.8): 3.9 ± 2.5 injections in the first year; and 2.5 ± 2.5 injections in the second year.

We analyzed the temporal profiles of CRT and BCVA throughout the course of two years (Figure 1). RMANOVA indicated that the CRT decreased (F (5, 460) = 23.8, *p* < 0.0001, η^2^ = 0.030) and BCVA improved (F (5, 430) = 14.2, *p* < 0.0001, η^2^ = 0.131) significantly through the observational period. The CRT significantly decreased at three months and thereafter until 24 months, as indicated by Steel’s multiple-comparison test (3 months; *p* < 0.0001, r = 0.69, 6 months; *p* < 0.0001, r = 0.81, 12 months; *p* < 0.0001, r = 0.77, 18 months; *p* < 0.0001, r = 0.77, 24 months; *p* < 0.0001, r = 0.84). The BCVA also improved significantly at three months and thereafter (3 months; *p* = 0.0069, r = 0.36, 6 months; *p* = 0.0002, r = 0.47, 12 months; *p* = 0.0046, r = 0.38, 18 months; *p* = 0.0013, r = 0.42, 24 moths; *p* = 0.0069 r = 0.37).

We analyzed the temporal profiles of HbA1c indicating glycemic control. HbA1c at baseline was 7.40 ± 1.08%, with no significant change over the two years (Figure 2A). The median value of HbA1c at baseline was 7.20%, and we divided the patients into a higher HbA1c group (≥7.2, *n* = 57) and a lower HbA1c group (<7.2, *n* = 55). RMANOVA showed that the change in HbA1c over time was statistically significant in both the high group (F(5, 155) = 4.75, *p* = 0.0046, η^2^ = 0.068) and the low group (F(5, 170) = 4.77, *p* = 0.0043, η^2^ = 0.048). In the higher group, average HbA1c at baseline was 8.23 ± 0.85 and significantly decreased to 7.69 ± 1.06 at 3 (*p* = 0.0367, r = 0.33), 12 (*p* = 0.0372, r = 0.33), and 18 (*p* = 0.0374, r = 0.33), as indicated by Steel’s multiple-comparison test (Figure 2B). In the lower group, HbA1c at baseline was 6.54 ± 0.44, showing no significant change over the two years. Only 6.25% (7/112) of patients had additional or intensified hypoglycemic medication within the two years.

The relationship between the changes in BCVA and HbA1c over the two-year period was analyzed (Figure 3). Proximity to the lower-left corner implies improvement in BCVA and a decrease in HbA1c over the two years. In all patients, no significant correlation was found. However, when the analysis was limited to 19 patients with worsening vision over the period, there was significant correlation between changes in HbA1c and BCVA (*p* = 0.0155, Y = 0.216 + 0.085X; R^2^ = 0.299). Robust regression analysis shows similar findings (*p* = 0.0488). Visual acuity worsened as HbA1c levels increased.

Regarding the results of the questionnaire survey (Table 2), for Question 1, 59.8% (67/112) of patients answered “yes” (Group A1a), while the remaining patients answered “no” (Group A1b). The HbA1c level in Group A1a was significantly lower than that in Group A1b at 6 (z = 2.82, *p* = 0.0047, r = 0.28), 12 (z = 3.60, *p* = 0.0003, r = 0.36) and 18 months (z = 2.52, *p* = 0.0117, r = 0.26), as indicated by the Mann–Whitney U test. There was no significant change in HbA1c at any time point in both groups.

For Question 2, 75.4% (89/112) of patients answered “yes” and were grouped into the A2a group. In both groups, HbA1c did not change significantly over the period, and no significant change was observed between the groups (Figure 4B).

For Question 3, the percentages of respondents in Group A3a (“Yes”) and Group A3b (“No”) were 66.1% (74/112) and 33.9% (38/112), respectively. HbA1c in either group did not change significantly. The value of HbA1c in Group A3a was significantly lower than that in Group A3b at 12 (z = 3.32, *p* = 0.0009, r = 0.33), 18 (z = 2.67, *p* = 0.0075, r = 0.27), and 24 months (z = 2.09, *p* = 0.0363, r = 0.21), as indicated by the Mann–Whitney U test, (Figure 4C). As shown in Table 2, 70.2% (52/74) of patients who had started regular ophthalmologic visits after initiating anti-VEGF therapy (Group A3a) responded in the questionnaire that they became more active with glycemic control (Group A1a). The percentage of A1a patients who had regular ophthalmology visits before starting anti-VEGF therapy (A3b) was significantly lower, at 39.5% (15/38) (*p* = 0.0016; chi-squared test).

For Question 4, 67.9% (76/112) of patients answered “yes” (Group A4a), while 32.1% (35/112) answered “no” (Group A4b). HbA1c in Group A4a was significantly lower than that in Group A4b (z = 2.10, *p* = 0.0351, r = 0.21) at 18 months, as indicated by the Mann–Whitney U test (Figure 4D). No significant change of HbA1c levels was noticed in either group.

In Japan, patients under 70 years old currently need to pay 30% of their own medical costs, while those over age 70 and 75 pay 20% and 10%, respectively. Even if the patient is over 70 years of age, if they have a higher income, they have a 30% cost. We examined the trend of HbA1c in patients with a 30% copayment (*n* = 76) and those with less than 30% copayment (*n* = 36) for anti-VEGF therapy (Figure 5A). Most patients (80.3%, 61/76) with a 30% copayment and 41.6% (15/36) of patients with less than 30% copayment responded that anti-VEGF treatment was expensive.

Of the 76 patients with 30% copayments, 15 were older than 70 years old and had higher incomes. To determine the effect of age, the trends of HbA1c were compared using 70 years as a threshold (Figure 5B). There was no significant change of HbA1c in either group, and the differences of HbA1c between the groups were insignificant.

## 4. Discussion

Active self-management by patients is necessary to prevent the worsening of DM and related complications [17]. DM is a lifestyle-related disease, and patients are instructed by physicians to try to control their glycemic levels through diet and exercise therapy. Pharmacological intervention has an effect on reducing the psychological stress of the patients and improve the glycemic control [18]. Similarly, our data show that the initiation of anti-VEGF therapy for DME affects HbA1c levels in relation to more aggressive glycemic control and higher medical costs. This result suggests that the initiation of anti-VEGF therapy may be an opportunity for patients to change their attitudes toward glycemic control. In our data, there was no significant overall change in HbA1c during the two-year anti-VEGF treatment period; however significant reductions were found in the group of patients with a baseline HbA1c of 7.2% or higher. It may have been difficult to further reduce the mild HbA1c levels at baseline. The group of patients who actively engaged in glycemic control via diet and exercise therapy after starting anti-VEGF therapy showed significantly greater reductions in HbA1c. In only 6.25% of patients, additional hypoglycemic medications were prescribed during the treatment period. Therefore, the decrease in HbA1c levels observed in this study could not be attributed to an increase in the dose of hypoglycemic agents. Additionally, anti-VEGF agents do not have the pharmacological effect of directly lowering HbA1c. The reason for the decrease in HbA1c is probably due to the contribution of the increased motivation of patients for glycemic control, including diet and exercise therapy. Exercise has clinical benefits, such as improved insulin sensitivity and reductions in HbA1c [19]. Moreover, Kuwata et al. showed that higher physical activity was associated with a lower incidence of DR in Japanese patients with Type 2 diabetes [20]. However, the ophthalmologist should be informed by the internist about the appropriate amount of exercise and dietary restrictions.

Center-involved DME, which can occur at any stage of DR, causes distorted vision and sudden vision loss. In the absence of DME, it is difficult to be aware of visual impairment at the nonproliferative DR stage. It is not rare for patients to realize the existence of ocular complications only after they notice vision loss due to DME. Our data show that 25.9% of patients did not know that DM can cause visual impairment until they became aware of their own impairment. Patients who are knowledgeable about diabetes [21], and have a good understanding of their status in glycemic control and complication risk [22] are better able to self-manage diabetes and control blood glucose. However, the presence or absence of knowledge that diabetes causes vision loss hardly influenced the change in HbA1c after the initiation of anti-VEGF therapy. Regardless of prior knowledge, it is important for ophthalmologists to thoroughly explain DME and anti-VEGF therapy to patients at the induction of treatment and to make sure of their understanding.

In our data, 74.1% of patients knew that diabetes might cause vision loss, but only 33.9% had regular visit to ophthalmologist before vision loss. Although regular eye examinations are recommended for diabetic patients, only 40–50% of DM patients receive fundus examinations at least once a year, which is not enough to manage DR [23]. To improve the quality of diabetic care, mutual collaboration among individual patients, ophthalmologists, and physicians is necessary. Patients who started regular visits to ophthalmologists after initiating anti-VEGF therapy showed significantly lower HbA1c at 12 and 18 months than that of patients who had previously had regular visits. Patients in Group A3a may have been less alarmed by DME-induced vision loss than patients in Group A3b were, and may have been more anxious and thus more aggressive in their glycemic control after they had experienced vision loss. Indeed, the ratio of patients who actively performed exercise and diet therapy was significantly higher in Group A3a than that in Group A3b; this may also be the reason Group A3a had lower HbA1c than that of Group A3b.

Anti-VEGF drugs are very expensive and represent a significant financial burden for patients. In a survey of ophthalmologists, the most common response was that the problem with anti-VEGF therapy was the financial burden on patients [24]. In our study, a high percentage of patients (67.9%) stated that the cost of anti-VEGF therapy was high. The HbA1c levels of these patients was significantly lower at 18 months compared to patients who did not consider anti-VEGF therapy to be expensive. The medical expenses of patients were determined by age and financial income; our data show that age and financial burden were not significant factors in the change in HbA1c. Anti-VEGF drugs need to be administered repeatedly, and more frequent injections mean an increased financial burden for patients. However, it is likely that the influence of the increased financial burden on glycemic control is small.

It is controversial whether a reduction in HbA1c contributes to the restoration of vision with anti-VEGF treatment. Bansal et al. showed that anti-VEGF treatment improves visual acuity regardless of HbA1c level [25]. On the other hand, Matsuda et al. reported that a lower HbA1c at the start of anti-VEGF therapy resulted in better visual acuity at one year [26]. In our analysis, there was no significant relationship between visual acuity and change in HbA1c over two years. Compared to the strong potential to improve the visual acuity of anti-VEGF drugs, the impact of a slight change in HbA1c may be small. In our study, only 19 patients (17.0%) had worsening visual acuity over the two years; as the level of HbA1c increased, visual acuity worsened. Poor glycemic control may be involved in the pathogenesis of aversion to anti-VEGF therapy. In support of this, groups with a poor response to anti-VEGF therapy had higher HbA1c and a greater foveal avascular zone than those of groups with good responses to anti-VEGF therapy [27]. Previous studies reported that the number of injections of anti-VEGF drugs for DME tend to decrease from year to year [28,29]. In anti-VEGF therapy with fewer injections, adequate glycemic control may play an important role in reducing the risk of edema recurrence. Long-term analyses are required to further understand the importance of glycemic control.

A periodic screening in patients with DR, and follow-up in patients with DME, are fundamental but are hindered by both economic reasons and distance, particularly in rural populations and during the COVID-19 pandemic [30]. Telemedicine-based medical care is one solution to these issues, and it can also confirm the motivation of patients to control their blood glucose [31]. Alternatively, increasing the frequency of blood glucose self-monitoring may help in optimizing the lifestyle and consequently improving HbA1c.

The limitation of this study is that the analysis was based on a subjective questionnaire; even if participants felt that they had become more aggressive in controlling their blood glucose, there was variation among patients. Another limitation is the limited number of the studied subjects. In addition, several factors other than hyperglycemia are involved in influencing DR and DME, including renal function, hypertension, and hyperlipidemia [32,33,34,35,36]. Although we focused on HbA1c in this study, further analyses with more cases are needed to understand the influence of these multiple factors.

This multicenter study shows that the initiation of anti-VEGF therapy for DME influences the change in HbA1c levels that are associated with increased motivation for glycemic control. Therefore, ophthalmologists should also consider patient psychological care after initiating anti-VEGF therapy.

## Figures and Tables

**Figure 1 jcm-11-04659-f001:**
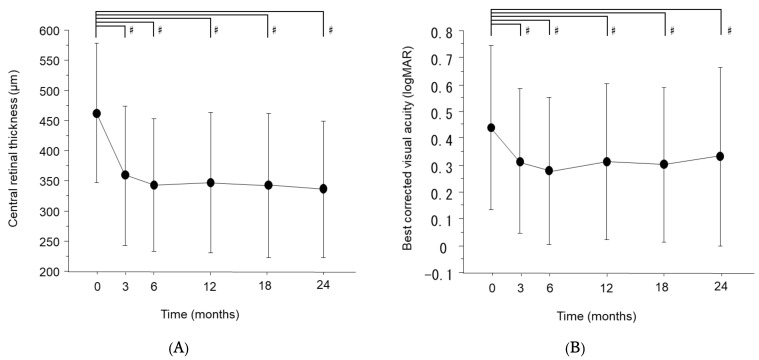
Changes in central retinal thickness (CRT) and best-corrected visual acuity (BCVA) after initiation of anti-vascular endothelial growth factor (VEGF) therapy. (**A**) CRT and (**B**) BCVA were measured at 0 (baseline), and 3, 6, 12, 18, and 24 months after initial injection of anti-VEGF agents. BCVA is expressed as the logarithm of the minimal angle of resolution (logMAR). Data are represented as means ± standard deviations (SD). ^#^
*p* < 0.05 (versus baseline by Steel’s multiple-comparison test).

**Figure 2 jcm-11-04659-f002:**
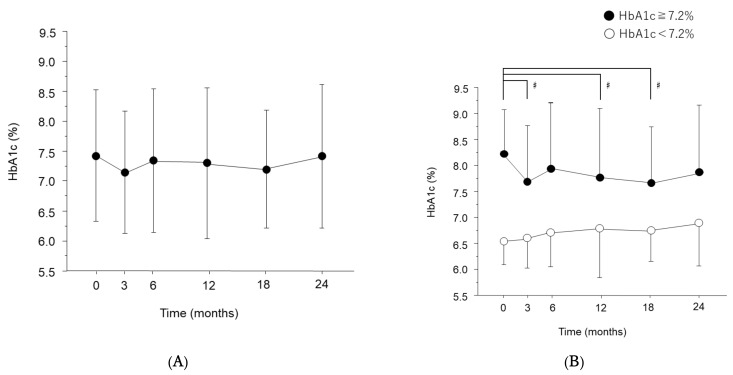
Time course of hemoglobin A1c (HbA1c) after initiation of anti-vascular endothelial growth factor (VEGF) therapy. The levels of HbA1c were measured in all eyes (**A**) and in patients with ≥7.2% (●), and <7.2% (○) HbA1c at baseline (**B**). Data are represented as means ± standard deviation (SD). # *p* < 0.05 (versus baseline by Steel’s multiple-comparison test).

**Figure 3 jcm-11-04659-f003:**
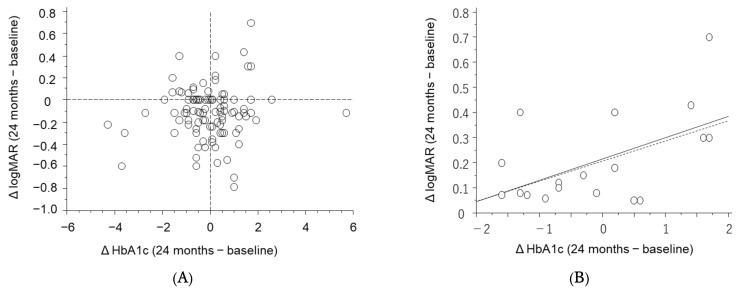
Linear correlation between changes in best-corrected visual acuity (BCVA) [logarithm of the minimal angle of resolution (logMAR)] and hemoglobin A1c (HbA1c) over the two-year period. (**A**) No significant correlation was found in all patients. (**B**) In patients with worsening BCVA, significant correlations were found (*p* = 0.0155, R^2^ = 0.299). Solid line: simple regression analysis. Broken line: robust regression analysis. The vertical dotted line indicates no change compared with baseline in HbA1c and BCVA values. Proximity to the lower-left corner implies improvement of BCVA and a decrease in HbA1c levels over the period.

**Figure 4 jcm-11-04659-f004:**
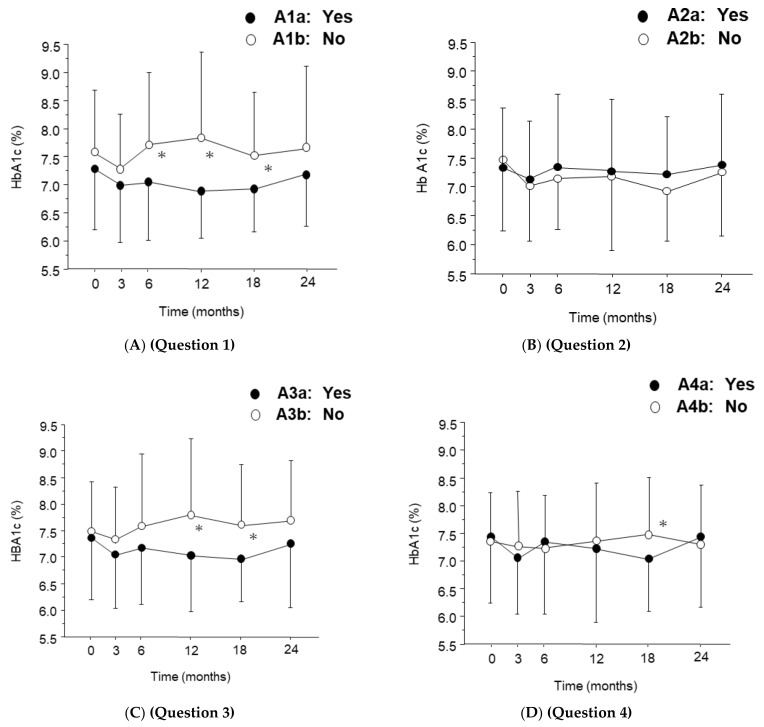
Temporal profile of hemoglobin A1c (HbA1c) in the groups according to the survey responses after initiation of anti-vascular endothelial growth factor (VEGF) therapy. HbA1c levels were measured in patients who answered yes (●), and no (○) to Questions 1–4. Data are represented by mean ± standard deviation (SD). * *p* < 0.05 (versus group by Mann–Whitney U test).

**Figure 5 jcm-11-04659-f005:**
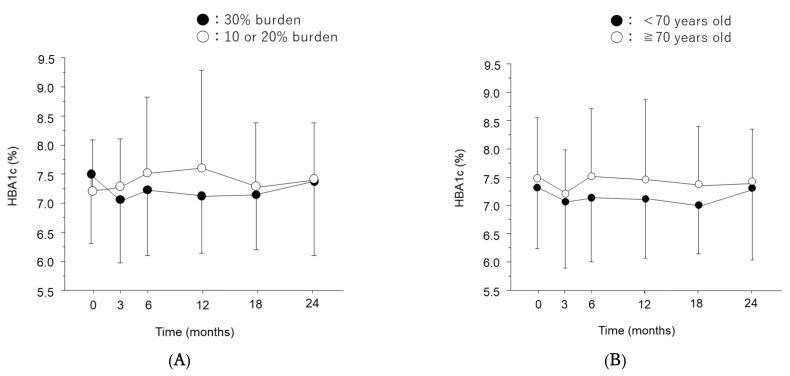
Time course of hemoglobin A1c (HbA1c) after initiation of anti-vascular endothelial growth factor (VEGF) therapy. HbA1c levels were measured in patients with a 30% copayment (●), and 10% or 20% copayment (○) (**A**); and in patients aged ≥70 years (●), and <70 years (○) (**B**). Data is represented as mean ± standard deviation (SD).

**Table 1 jcm-11-04659-t001:** Baseline characteristics of enrolled patients.

	Patients (*n* = 112)
Age (year)	67.0 ± 10.3
Male gender	80/32 (71.4%)
Duration of diabetes mellitus (year)	12.7 ± 10.6
Number of anti-VEGF injections (2 years)	6.4 ± 4.5
Hemoglobin A1c (%)	7.4 ± 1.08
Creatinine (mg/dL)	1.04 ± 1.16

**Table 2 jcm-11-04659-t002:** Questionnaire structure and answer distribution.

Question	Groups	*n* (%)	The Ratio of A1a *n* (%)	*p* Value ^a^
1. Since starting anti-vascular endothelial growth factor (VEGF) therapy, have you become more active in glycemic control through diet and/or exercise therapy?	A1a: Yes	67 (59.8)	67/67 (100)	-
	A1b: No	45 (40.2)	0/45 (0)	
2. Until you were aware of your vision impairment due to DME, did you know that diabetes mellitus can cause visual impairment?	A2a: Yes	83 (74.1)	47/83 (56.6)	0.2433
	A2b: No	29 (25.9)	20/29 (69.0)	
3. Have you started to regular visit an ophthalmologist after vision loss?	A3a: Yes	74 (66.1)	52/74 (70.2)	0.0016
	A3b: No	38 (33.9)	15/38 (39.5)	
4. Do you think the cost of anti-VEGF drugs is high?	A4a: Yes	76 (67.9)	47/76 (61.8)	0.5262
	A4b: No	36 (32.1)	20/36 (55.6)	

^a^ Chi-squared test.

## Data Availability

The datasets generated during and/or analyzed during the current study are available from the corresponding author on reasonable request.

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
