# Peer review of "Glycemic Control after Initiation of Anti-VEGF Treatment for Diabetic Macular Edema"

_jcm, 2022, doi:10.3390/jcm11164659_

Round 1

Reviewer 1 Report

The authors did not record the results of all statistical tests according to scientific standards (e.g., RMANOVA).

“Recent randomized clinical trials show that anti-VEGF therapy improves visual acuity and macular swelling [6–9].” - too broadly described.

For the variance analysis, the effect size would also need to be calculated and interpreted (e.g. eta-square). This would show the significance of the obtained results.

Not only the main exclusion criteria should be described, but the others.

Reviewer 2 Report

In this interesting retrospective multicenter study, the authors observed an improvement in HbA1c 6 and 18 months after starting anti-VEGF therapy. However, this reviewer raises some issues that need to be addressed.

The main limitation of the study, as pointed out by the authors, is the evaluation based on a questionnaire. However, there are other issues the authors need to address:

1- Were there any changes in antihyperglycemic therapy that could have influenced the improvement in HbA1c?

2- Could the frequency of glycemic self-monitoring be increased in this period and in this way have conditioned an optimization of at least the lifestyle (diet and patient confidence) with consequent improvement of HbA1c?

Another limitation is the limited number of subjects studied.

Moreover, in times of the COVID-19 pandemic, the possibility of diagnosing DR by telemedicine (1- Diabetes Metab Res Rev. 2019 Mar;35(3):e3113. doi: 10.1002/dmrr.3113. 2- J Diabetes Res. 2020 Oct 14;2020:9036847. doi: 10.1155/2020/9036847.), could be an important solution in geographical areas where the movements of patients to specialized centers can be long and demanding. Certainly, this issue and above references should be commented in discussion.

Round 2

Reviewer 2 Report

No further comments.

This manuscript is a resubmission of an earlier submission. The following is a list of the peer review reports and author responses from that submission.

Round 1

Reviewer 1 Report

The introduction is too sketchy. It does not inform about the research carried out so far (it does not reflect the topic of the article).

There are no bioethical commission approval numbers.

When making so many comparisons between the means, the statistical tests used are completely unjustified.

The results of statistical tests are not recorded according to scientific standards, e.g. Z = 23; p = 0.02 or U = 89; p = 0.46.

The authors did not calculate the effect size for the statistical tests used. The p-value alone is definitely not enough.

The assumptions of the regression analysis (homoscedasticity, etc.) have not been analyzed. For example, outliers may influence the obtained results.

Authors should use more advanced statistical tests.

The discussion does not relate to the existing literature. It is not a discussion of the obtained results at all.

Reviewer 2 Report

In this retrospective multicenter study, patients treated with anti-VEGF therapy for diabetic macular edema showed a reduction in HbA1c. Decrease in HbA1c is probably due to the contribution of increased motivation of patients for glycemic control. Therefore psychological aspects linked both to the awareness of retinal complication and to economic aspects can influence the lifestyle of patients, with an impact on glycemic control.

The manuscript is interesting. The limitations of the study are correctly reported by the authors themselves. The conclusions are supported by the results. This review suggests a few issues that the authors should address to enrich the discussion.

1- Diabetic retinopathy, including maculopathy, is one of the main causes of blindness in the adult population. In particular, it is the first cause in Western countries. The possibility of performing a periodic screening in patients without known maculopathy, and a periodic follow-up in patients with a known diagnosis, is fundamental but is made particularly difficult both for economic reasons and for distance, particularly in rural populations. Therefore, this is one of the main challenges for ophthalmologists. Therefore, possibility of studying fondus oculi through telemedicine is certainly to be considered, especially as a screening method for outpatients and in particular during the current pandemic (Diabetes Metab Res Rev. 2019 Mar;35(3):e3113. doi: 10.1002/dmrr.3113. - J Diabetes Res. 2020 Oct 14;2020:9036847. doi: 10.1155/2020/9036847). This important issue and above references need to be discussed by the authors.

2- A table with the clinical characteristics of the patients (duration of diabetes, age, HbA1c, gender, etc.) would be useful for the reader.

3- In the text there are some typos.

Round 2

Reviewer 1 Report

Statistical analysis still raises a lot of doubts.

In the manuscript, the authors did not include the recorded results of the analysis of variance (according to scientific standards).

It is still unknown in the manuscript where and what statistical test was used.

A detailed justification must be provided for when the Mann-Whitney U test and when the ANOVA was used.

In the case of multiple measurements in the same group of patients, statistical tests appropriate for the repeated measurements should be used.
